# Highly Stretchable and Sensitive Multimodal Tactile Sensor Based on Conductive Rubber Composites to Monitor Pressure and Temperature

**DOI:** 10.3390/polym14071294

**Published:** 2022-03-23

**Authors:** Bing Zhu, Chi Ma, Zhihui Qian, Lei Ren, Hengyi Yuan

**Affiliations:** 1State Key Laboratory of Automotive Simulation and Control, Jilin University, Changchun 130025, China; zhubing@jlu.edu.cn (B.Z.); machi20@mails.jlu.edu.cn (C.M.); 2Key Laboratory of Bionic Engineering, Jilin University, Changchun 130025, China; yuanhy18@mails.jlu.edu.cn; 3School of Mechanical, Aerospace and Civil Engineering, University of Manchester, Manchester M13 9PL, UK

**Keywords:** multimodal sensors, stretchable tactile sensors, resistance-type sensors, conductive rubber, carbon nanomaterials

## Abstract

Stretchable and flexible tactile sensors have been extensively investigated for a variety of applications due to their outstanding sensitivity, flexibility, and biocompatibility compared with conventional tactile sensors. However, implementing stretchable multimodal sensors with high performance is still a challenge. In this study, a stretchable multimodal tactile sensor based on conductive rubber composites was fabricated. Because of the pressure-sensitive and temperature-sensitive effects of the conductive rubber composites, the developed sensor can simultaneously measure pressure and temperature, and the sensor presented high sensitivity (0.01171 kPa^−1^ and 2.46–30.56%/°C) over a wide sensing range (0–110 kPa and 30–90 °C). The sensor also exhibited outstanding performance in terms of processability, stretchability, and repeatability. Furthermore, the fabricated stretchable multimodal tactile sensor did not require complex signal processing or a transmission circuit system. The strategy for stacking and layering conductive rubber composites of this work may supply a new idea for building multifunctional sensor-based electronics.

## 1. Introduction

Stretchable and flexible tactile sensors for direct human body contact applications are attracting increasing interest in academic and industry fields. With the development of stretchable and flexible electronics, high-performance stretchable tactile sensors play an increasingly important role in a variety of applications including human motion detection, stretchable smart robots, human-machine interaction, and wearable medical devices [1,2,3,4]. 

Numerous studies have reported on the development and fabrication of highly stretchable and sensitive tactile sensors. According to their working mechanism, tactile sensors include resistance [5], capacitance [6,7], piezoelectric, and triboelectric-type sensors [8]. To improve stretchable tactile sensor performance, researchers have enhanced their structures and morphologies [9,10,11,12], and prepared stretchable tactile sensors or sensor arrays [13,14,15] via complex microelectronic processes [16,17,18] and bionic methods [19,20]. For example, Chen et al. created a graphene-based resistive strain sensor with a sensitivity of 20.1 [5], and Gao et al. fabricated a capacitance-type sensor based on conductive silicone rubber and pure PDMS that can detect dropping water, feet lifting, and walking [6]. Pyo et al. fabricated a pressure sensor with a multi-layered structure and high sensitivity (26.13 kPa^−1^) over a wide pressure range (0.2–982 kPa) [10]. Jian et al. also fabricated a pressure sensor with high sensitivity (19.8 kPa^−1^, <0.3 kPa), a low detection limit (0.6 Pa), and a fast response time (<16.7 ms) due to its bionic hierarchical structures [19].

Tactile properties require a combination of roughness, hardness, thermal conductivity, temperature, humidity, sharpness, vibration, force, and other parameters in human skin through contact with surface objects [21,22,23], thus the development of multifunctional or multimodal tactile sensors has gained interest worldwide [24,25,26,27,28]. Unlike human skin, most stretchable sensors can only effectively sense one parameter, and researchers are investigating stretchable tactile sensors with improved stretchability, ductility, and sensitivity and measurement ranges. In addition, many conductive composite materials used as stretchable tactile sensors are sensitive to one type of tactile stimulus (mostly pressure or strain), as well as other parameters such as temperature and humidity [29,30,31]. Efforts were already made to engineer stretchable sensors that can detect various parameters simultaneously. For example, Liu et al. reported a stretchable multimodal sensor that can detect multiple stimuli with only one device by spraying a mixture of carbon black (CB) and reduced graphene oxide (rGO) on a paper substrate [32]. However, it was troublesome for the sensors in this report to distinguish different stimuli. Jung et al. also proposed a stretchable device containing pressure, temperature, and piezoresistive hair-type flow sensors, the latter of which was fabricated with a mixture of carbon nanotubes and polydimethylsiloxane piezoresistive materials [27]. However, the sensor manufacturing process in the study was complicated because each sensitive layer was prepared separately, and the output signals were diverse. Current multimodal sensors are limited by complex manufacturing processes, high cost, and complex signal processing. Therefore, challenges remain to implementing high performance multimodal sensors. 

In this study, we developed a new stretchable tactile sensor consisting of conductive rubber composites with improved sensitivity, measurement range, and measurement functionalities. Due to its unique advantages, such as good conductivity, high chemical and thermal stability and low toxicity, carbon materials have great application potential in the field of stretchable and flexible sensors. For example, CNT has good conductivity, high aspect ratio and excellent flexibility. In addition, GP also has excellent flexibility and good conductivity, while CB has the advantages of low cost and good conductivity. Therefore, different conductive rubber composites were prepared with these carbon nanomaterials as conductive fillers in this work. Because rubber materials have good stretchability and biocompatibility, we constructed stretchable tactile sensors with silicone rubber materials and carbon nanomaterials [33,34,35]. The conductive rubber composites were prepared with carbon nanomaterials (multiwalled carbon nanotubes (MWCNTs), carbon black (CB), and graphene (GP)) incorporated into room temperature vulcanized silicone rubber (RTV). After testing the reinforcement of the conductive fillers on the mechanical and electrical properties of the conductive rubber composites, the stretchable tactile sensors were fabricated with a conductive rubber composite film as the sensitive layer. Then, an appropriate material ratio was selected according to the experimental data to properly integrate the stretchable multimodal tactile sensor. The fabrication method used in this study was simple and inexpensive. Based on the pressure-sensitive and temperature-sensitive effects of the conductive rubber composites, a final stretchable multimodal tactile sensor with a dual-sensitive layer was designed and fabricated. This sensor, with a simple structure and manufacturing process can simultaneously measure pressure and temperature. Since there are two signal outputs, the signal processing of this sensor was simple and did not interfere with each other. In addition, the stretchable multimodal tactile sensor developed in the study showed good sensitivity when exposed to compression at different temperatures.

## 2. Experimental Detail

### 2.1. Materials

Superconducting carbon black (CB) powder (CABOT BP2000) was purchased from Hefei Saibo New Materials Co., Ltd., Hefei, China. Multilayer graphene (GP) (5–10 layers, 95% purity) and multi-wall carbon nanotubes (MWCNTs) (TF-25001, inner diameter 3–5 nm, outer diameter 8–18 nm, length 3–12 μm, 95% purity) were obtained from Suzhou carbon graphene Technology Co., Ltd., Suzhou, China and were used as conductive fillers to fabricate the conductive rubber composites. Room temperature vulcanized silicone rubber (RTV) (viscosity: 10,000 mps) and its curing agent were purchased from Jinan Guobang Chemical Co., Ltd., GB-107, Jinan, China. The RTV was subsequently used as the matrix material. A conductive paint pen (6 mL) whose volume resistance is 2.5 mΩ/mm^2^ and curing time is 24 h was purchased from Shenzhen Xinwei New Material Co., Ltd., Shenzhen, China. PET tape which was used as a stretchable electrode was purchased from Shenzhen Ausbond Co, Ltd., Shenzhen, China. High temperature resistant PET tape (thickness 60 μm) was used as a substrate to carry the sensors, and fiberglass adhesive tape was used as an insulating material to encapsulate the sensors.

### 2.2. Equipment

An analytical balance (ME104E/02, Mettler Toledo instruments (Shanghai) Co., Ltd., Shanghai, China) was used to weigh the experimental materials. The conductive fillers were dispersed in anhydrous ethanol using an ultrasonic disperser (FS-100t, Shanghai Shengxi ultrasonic Co., Ltd., Shanghai, China), and the conductive fillers, curing agent, and rubber matrix were evenly mixed with a digital display electric mixer (JB300-SH, Shanghai Specimen and Model Factory, Shanghai, China). After mixing, the conductive rubber composites were vacuumed with a vacuum pump (V-I140SV, Zhejiang Value Mechanical and Electrical Products Co., Ltd., Wenling, China). Then, the conductive rubber composites were spin-coated on the surface of a culture dish with a spin coater (12 A, Zhangqiu Guanpai Electronics Co., Ltd., Zhangqiu, China). A vacuum drying oven (DZF6050, Shanghai YIHENG Technical Co., Ltd., Shanghai, China) was used to provide a curing temperature environment for the conductive rubber composites. A DC power supply (IT6411, Taiwan ADEX Electronics Co., Ltd., Nanjing, China) was used to test the volt-ampere characteristics of the conductive rubber composites. In addition, a desktop digital multimeter (34465 A, Keysight Technologies Co., Ltd., Beijing, China) was used to test the electrical signals of the conductive rubber composites. An electronic universal testing machine (ZQ-990, Zhiqu Precision Instruments Co., Ltd., Dongguan, China) was used for the pressure and stretch test (Appendix A), and a constant temperature heating table with a double digital display (JF-965S, Dongguan Jinfeng Electronics Co., Ltd., Dongguan, China) was used to provide the proper temperature environment. Lastly, a scanning electron microscope (SEM) (S-4800, Hitachi, Ltd., Tokyo, Japan) was used to observe the microstructures of the conductive rubber composites. 

### 2.3. Preparation of the Conductive Rubber Composites

The fabrication process of conductive rubber composite film was shown in Figure 1a. The materials were prepared according to the following steps. First, the weighed conductive fillers were stirred and dispersed in ethanol with a glass rod, the volume of ethanol was 30% of that of the rubber matrix material. Then the conductive fillers were dispersed in an ultrasonic disperser for 1 h. Next, RTV matrix proportional to the conductive filler was added to the beaker and the mixed material was stirred at 300 rpm using an electric stirrer for 30 min. Subsequently, a curing agent with a mass fraction of 5% of the RTV matrix was added and stirred with an electric stirrer at 500 rpm for 30 min. Next, a vacuum pump was used to continuously remove any bubbles at a vacuum degree of −0.09 MPa for 30 min. Then, the liquid mixture was removed with a needle tube and injected into a polytetrafluoroethylene mold or PS culture dish, and then the mixed liquid materials were spin-coated at a speed of 200 rpm for 120 s. Finally, the mixtures were placed in a vacuum drying oven and maintained at 60 °C for 24 h. 

Three types of carbon nanomaterials, including zero-dimensional (0-D) CB, 1-D CNT, and 2-D GP, were used as the conductive fillers. Combining various conductive fillers was advantageous, as the percolation threshold of the composite conductive fillers could be effectively reduced [36]. A pre-experiment was conducted to determine the mass ratio of the series of conductive rubber composites, and the quality of conductive filler is 8% of that of RTV matrix. For the conductive rubber composites containing hybrid fillers of CB/GP or CB/CNT, the mass ratio of total fillers remained unchanged, and the selected mass ratio of CB were: CNT or GP = 1:1, CB: CNT or GP = 2:1, CB: CNT or GP = 3:1. Lastly, seven groups of conductive rubber composites were successfully prepared and tested.

### 2.4. Preparation of the Stretchable Tactile Sensor

The stretchable compression and temperature-sensitive sensor developed in this study was assembled layer by layer, and the structure of the stretchable tactile sensor is shown in Figure 1b, while Figure 1c shows a photograph of the sensor. The preparation process of the stretchable compression-sensitive and temperature-sensitive sensor was as follows. First, two parallel conductive paths were drawn on the sticky side of the PET tape using a conductive copper paint pen. Second, 30 × 30 mm conductive composite films were cut and transferred to the side of the PET substrates coated with electrodes. Thirdly, the unfinished sensors were cured in a vacuum oven at 60 °C for 2 h, and then the electronic wires were welded on the cured electrode for testing. Finally, surface packaging was fabricated with fiberglass tape. The stretchable stretch-sensitive sensor was assembled in a similar manner. However, to prevent any effects on the stretch test, the PET tape was only used as the substrate in sections containing electrodes. In addition, the shape of the conductive composite film that was applied to stretch-sensitive sensor was modified, from 30 × 30 × 1 mm to 10 × 50 × 1 mm to obtain a wider deformation range during stretch test. After testing different conductive rubber composites, a stretchable multimodal tactile sensor with a dual-sensitive layer was assembled. The two conductive rubber composites with temperature and compression sensitivity properties were stacked together to provide mutual insulation and an orthogonal arrangement of the electrodes. This uncomplicated method guaranteed that the input signals were unified while the output signals were processed separately.

### 2.5. Characterization of the Stretchable Tactile Sensor

The volt-ampere characteristics of sevens conductive composite films with the same shape (10 × 50 × 1 mm) were tested from 0 to 9 V with a DC power supply. The conductivities of conductive rubber composites can be calculated from the volt-ampere characteristic curves.

The mechanical properties were tested using an electronic universal testing machine using dog-bone conductive rubber composites. These samples were prepared with NC machined polytetrafluoroethylene mold to meet the testing standards, and elastic modulus, tensile strength, and elongation at break were measured. Following the GB/T528-2009 standard, the cross-sectional area of the specimens was 4 × 2 mm, with a gauge distance of 25 mm [37,38,39]. The moving speed of the fixture in the universal testing machine was 50 mm/min until the specimens broke. During the stretch test, three specimens of each conductive rubber composite type were used to ensure the repeatability of the test results. The test devices mainly included loading equipment, a desktop digital multimeter, and a PC. The loading equipment consisted of an electronic universal testing machine and a constant temperature heating table with a double digital display. The electronic universal testing machine was used to obtain the stretch and compressive test results, by replacing the fixture, and the performance of the stretch-sensitive and the compression-sensitive sensor was tested with a desktop digital multimeter and the electronic universal testing machine. To study the response of the stretchable tactile sensor to pulling and pressures, the probes of the digital multimeter were fixed on the universal testing machine fixture, so that they could move synchronously. In the stretch test, both ends of the stretchable tactile sensor were clamped in the jaws of the upper and lower fixtures. In the compression test, the sensor was fixed on the lower round flat top portion of the electronic universal testing machine fixture. The electronic wires welded on the electrodes were then connected with the probes from a digital multimeter. The four-wire resistance method was used to eliminate the effects of contact resistance. When applying compression or stretch to the sensor, the resistance changes in the sensor were recorded by the digital multimeter. During the stretch test, the loading speed of the stretch stroke was 2 mm/min [37], and when the strain approached the elongation at break, the fixture returned automatically at a speed of 5 mm/min. During compression testing, the loading speed of the compression stroke was 0.5 mm/min [38], and when the pressure was greater than 110 kPa, the fixture returned automatically at a speed of 100 mm/min. The sampling interval of the desktop digital multimeter was 0.01 s.

For the temperature-sensitive property measurements, the sensor was fixed to the surface of the constant temperature heating table, and sensor resistance changes were recorded by the digital multimeter. Starting from 30 °C, the temperature varied at intervals of 5 °C [32]. The resistance was tested three times and the average resistance was recorded, and the sampling interval of the desktop digital multimeter was 0.01 s. The cross-sectional morphologies of the conductive rubber composites were observed with a scanning electron microscope. To inspect the dispersion quality of the conductive fillers, the cross-sections of the conductive composites were magnified by 20,000× and 50,000× at 3–5 kV accelerating voltage.

## 3. Results and Discussion

### 3.1. Morphology of Conductive Rubber Composites

Figure 2 depicts the cross-section scanning electron microscopy (SEM) micrographs of the conductive rubber composites, showing that the conductive fillers were well-dispersed in the RTV matrix and formed conductive pathways which, indicated by blue arrows in the Figure 2a,b, show the SEM micrographs of the pure silicone rubber and conductive silicone rubber filled with CB, respectively. Figure 2c,d shows the SEM micrographs of the conductive rubber composites containing the CB/CNT and CB/GP hybrid fillers, respectively. 

### 3.2. Mechanical Properties and Conductivities of Conductive Rubber Composites

The volt-ampere characteristics of the seven conductive rubber composites are shown in Figure 3a. According to a previous study [36], the percolation threshold of the carbon-based conductive filler was approximately 5–8 wt%. The rubber transformed into the conductive rubber composites when the conductive filler concentration exceeded the percolation threshold, and the conductive fillers overlapped and made contact with each other throughout the matrix [40,41,42]. 

The experimental results showed that CNT greatly improved the conductivity of the CB-embedded composites. When the mass ratio of CB and CNT was 1:1, the conductivity was more than 20 times that of the non-CNT-embedded composites. This was due to the high aspect ratio of CNT, which could effectively form overlapping conductive paths, and led to the synergistic effect of high conductivity at a low mass fraction of conductive fillers [41]. However, the embedded GP did not significantly affect the conductivity of the CB-embedded composites, which was possibly caused by the poor conductivity of GP embedded in the RTV matrix, and it is less able to disperse in the material compared with CNT [42].

The elastic modulus, tensile strength, and elongation at break of the conductive rubber composites with various mass ratios are shown in Figure 3b. Compared with the original rubber material, the tensile strength and elastic modulus of the conductive rubber composites increased with the introduction of CB. In addition, the tensile strength and elastic modulus of the conductive rubber composites further improved with the continuous addition of CNT. The tensile strength and elastic modulus increased by about 50% at the peak (CB:CNT = 3:1), and the elongation at break of the CB-embedded composites increased by 50% when the mass ratio of CNT and CB was 1:1. With continuous addition of GP, the elastic modulus decreased by 35% (CB:GP = 1:1); however, the tensile strength and elongation at break increased by 75% and 200% at the peak (CB:GP = 3:1), respectively. The high tensile strength was attributed to the reinforcement of the conductive fillers, which effectively transferred the load between the carbon nanomaterials.

L. Valentini et al. prepared ethylene-propylene-diene terpolymer rubber (EPDM)-based nanocomposites containing carbon black (CB), graphene nanoplatelets (GNPs), and mixtures of the two fillers [43]. They found that CB or silica when added to elastomers create a modulus that increases with strain. Besides, the sample EPDM-6 (i.e., 2 wt% of GNPs and 24 wt% of CB) showed a higher increment of the maximum strength along with a higher elongation at break with respect to the EPDM/CB blends. Unlike this study, the elongation at break of the sample EPDM-6 (i.e., 2 wt% of GNPs and 24 wt% of CB) is less than the sample EPDM without any fillers. This may relate to the fact that too much filler affects the good flexibility of EPDM. In another study, Md Najib Alam et al. explored the dispersion and reinforcement performance of binary fillers in natural rubber [44]. They found that binary fillers with a 1:1 ratio of silica to graphite powder provide excellent mechanical performance. Moreover, thermal oxidative aging resistance properties compared with single fillers and the binary filler system at only 20 phr of filler content shows an improved modulus and stress-at-break of approximately 110% and 15%, respectively, compared with the unfilled rubber vulcanizate. Their study also shows that the synergistic effect brought by various fillers is worthy of attention.

### 3.3. Characterization of Stretchable Tactile Sensors

#### 3.3.1. Stretch and Compression-Sensitive Properties

Lastly, the stretch-sensitive and compression-sensitive characteristics of the materials were tested. Relative changes in resistance (Δ*R*/*R*_0_) were obtained as the output signal of the sensor, and the stimulus was the pressure or strain provided by the universal testing machine. The gauge factor (GF) was used to evaluate the sensitivity, as it is considered one of the most important sensor performance parameters. The GF of stretch-sensitive characteristic was defined as: GF = (Δ*R*/*R*_0_)/Δ*ε*, and the GF of compression -sensitive characteristic was defined as: GF = (Δ*R*/*R*_0_)/Δ*P*, where Δ*R*/*R*_0_ is the real-time relative change in resistance, *R*_0_ is the initial resistance, *R* is the real-time resistance, *ε* is the strain, and *P* is the pressure [5]. The experimental stretch and compression sensitivity results of the fabricated sensors are shown in Figure 4. 

Figure 4a shows the relative resistance changes due to stretch. According to the stretch test results, GF showed a sharp increase with increased strain, which indicated that the sensitivity was dependent on strain. It is found that the sensitivity was higher when the strain was higher. This was attributed to the conductive paths, which were significantly reduced with maximum strain and when cracks appeared in the material [45]. With increased strain, the possibility of irreversible damage also increased. Therefore, GF was calculated in different strain ranges with good linearity. For example, the GF of 4CB4CNT-RTV was 37.45 (0–12%) and 247.54 (12–20%), and the linear fitting results of 4CB4GP-RTV showed that the GF in the linear range was 12.03 (0–20%) and 81.68 (20–35%). The GF of 5.33CB2.67CNT-RTV was 14.73 (0–20%) and 84.56 (20–30%), and the GF of 5.33CB2.67GP-RTV was 20.39 (0–15%) and 115.99 (15–20%). The GF of 6CB2CNT-RTV was 4.54 (0–15%) and 13.65 (15–25%), the GF of 6CB2GP-RTV was 25.11 (0–10%) and 129.81 (10–20%), and the GF of 8CB-RTV was 2.63 (0–40%). In conclusion, the GF values of the stretchable tactile sensors prepared in this study were 2–40 under low strain and 80–250 under high strain, and these values were similar to the reported results [45,46].

Figure 4b shows the pressure response curves of the sensors. The results of the compression test showed that most sensors exhibited excellent linearity. As shown in Figure 4b, the relative change in sensor resistance increased when exposed to pressure. According to the test results, the fabricated sensors exhibited excellent linearity for most samples. The GF of sensors based on different materials was also calculated. The data of 4CB4CNT-RTV showed that the GF in the linear range was 0.00204 kPa^−1^. The GF of 4CB4GP-RTV was 0.00577 kPa^−1^. The GF of 5.33CB2.67CNT-RTV was 0.00 401 kPa^−1^, and the GF of 5.33CB2.67GP-RTV was 0.01097 kPa^−1^. In addition, the GF of 6CB2CNT-RTV was 0.01171 kPa^−1^, the GF of 6CB2GP-RTV was 0.01161 kPa^−1^, and the GF of 8CB-RTV was 0.00803 kPa^−1^. In conclusion, the GF values of the conductive rubber composite stretchable tactile sensors under compression were 0.00204–0.01171 kPa^−1^, and these values were higher than the reported results [47,48].

#### 3.3.2. Temperature-Sensitive Properties

Temperature is also important in tactile property and temperature-sensing is essential for many systems. The temperature test results from this study are shown in Figure 4c. Similarly, the GF for temperature sensitivity was defined as the ratio of relative change in resistance to the temperature: GF = (Δ*R*/*R*_0_)/Δ*T*, where Δ*R*/*R*_0_ is the relative change in resistance, *R*_0_ is the initial resistance, *R* is the real-time resistance, and *T* is the temperature [49,50]. The resistance of the stretchable tactile sensor changed with increasing temperature according to the temperature resistance data. In this study, 4CB4GP-RTV and 5.33CB2.67GP-RTV exhibited higher sensitivities according to the test results. The GF of 4CB4GP-RTV was 2.92%/°C (30–50 °C), 11.39%/°C (50–70 °C), and 19.14%/°C (70–90 °C). The GF of 5.33CB2.67GP-RTV was 1.39%/°C (30–50 °C), 5.62%/°C (50–70 °C) and 19.91%/°C (70–90 °C), which was higher than commercial platinum temperature sensors (0.39%/°C). The test results showed that the temperature sensors fabricated in this study had excellent temperature-sensitive characteristics, and the GF values of the CB-embedded conductive rubber composite were 0.89%/°C (30–50 °C), 3.06%/°C (50–70 °C), and 13.92%/°C (70–90 °C). In contrast, the temperature sensitivity of the conductive rubber composites containing hybrid CNT/CB fillers was low. These results showed that the temperature sensitivity of the conductive rubber composites with synergistic effects should be considered according to different situations. Thus, embedded CNTs can inhibit the positive temperature effect of conductive rubber composites, and an increasing quantity of CNTs causes a more obvious effect. However, embedded GP can enhance the positive temperature effect of the conductive rubber composites. Thus, with increasing GP mass ratio, the positive temperature effect of the conductive rubber composites significantly improved.

This work showed that the sensitive characteristics of the conductive rubber composites embedded with hybrid conductive fillers differed from those only embedded with CB. In addition, compared with the CB-embedded conductive rubber composites, the CNT/CB-embedded and GP/CB-embedded conductive rubber composites had higher GF values and a wider linear range in stretch and compression tests. The excellent sensitive characteristics of the conductive rubber composites containing hybrid conductive fillers may thus be related to the improved mechanical properties and observed synergistic effects. 

In general, the results showed that the synergistic effect significantly impacted the sensitive characteristics of the conductive rubber composites, which can benefit the design and preparation of stretchable multimodal tactile sensors. The results also showed that the sensitivity and linearity of 5.33CB2.67GP-RTV were excellent, especially in the stretch, compression, and temperature tests. However, 6CB2CNT-RTV only exhibited better sensitivity characteristics in compression. To differentiate the functionalities of the two sensitive layers, these two materials were further studied in subsequent experiments. 

The tensile, compression and temperature sensitivities of conductive rubber composites with different material ratios in the work are summarized in Table 1.

#### 3.3.3. Sensing Properties Stability

The repeatability and reproducibility of stretchable tactile sensors are important factors that need to be considered for practical use. As shown in Figure 5, the stretchable 5.33CB2.67GP-RTV and 6CB2CNT-RTV tactile sensors were tested 500 times under stretch and 1500 times under compression cyclic loading-unloading [51,52]. The maximum strain load was 30% and the maximum pressure load was 50 kPa. Additionally, the speed of the stretch and compression strokes were 50 and 5 mm/min, respectively, and the output signal was the resistance of the sensors. When subjected to cyclic loading-unloading, the stretchable tactile sensors showed a drift in electrical signals. Specifically, the average resistance of sensors decreased with an increasing number of stretch loading-unloading cycles, and increased with increasing compressive loading-unloading cycling. This phenomenon was more noticeable in the curves for the 5.33CB2.67GP-RTV-based sensors. The results also indicated that the fatigue properties of the CNT-embedded conductive rubber composites were better than the GP-embedded conductive rubber composites. Thus, the sensors showed good repeatability and few variations after 1000 loading-unloading cycles.

#### 3.3.4. Sensing Mechanism

The cause of conductive rubber composites resistance changes under external stimuli has been studied by researchers [53,54,55], and resistance increases with loading and decreases with unloading proportionally, which is known as the positive force effect. The positive force effect occurs due to the disassembly and reassembly of the conductive paths. When an external force is applied, the conductive paths is destroyed due to the deformation of the conductive rubber composites. However, when the applied load is removed, the conductive rubber composites recover their shape and the conductive paths reassemble. We observed that the resistance of the sensors fabricated in this work increased with increasing stretch and compression, indicating that the results of the study followed this principle.

The influence of temperature on the conductive rubber composites was divided into positive temperature and negative temperature effects. The phenomenon where conductive rubber composites resistance increases with rising temperature is called the positive temperature effect. This is in contrast to the phenomenon known as the negative temperature effect. The mechanism of the positive temperature effect is due to the increase in temperature, where the polymer crystalline phase melts and expands, which increases the distance between the conductive filler particles. Hence, the resistance of the conductive rubber composites increased. The negative temperature effect occurred due to the increase in electron transitions with increasing temperature. According to the literature, the type and quantity of conductive fillers will significantly affect the dominant temperature effect in the conductive rubber composites [55]. Most sensors fabricated in this paper exhibited the positive temperature effect, which may be due to the high concentration of conductive particles. Moreover, adding GP to CB-embedded rubber composites leads to a more obvious temperature effect, while adding CNT to CB-embedded rubber composites leads to the opposite result. The experimental results in this paper verify the principle of temperature effect.

### 3.4. Preparation and Testing of the Stretchable Multimodal Tactile Sensor

According to the experimental results, the mass ratio of the material affects the sensitive characteristics of the conductive rubber composites. In this study, 5.33CB2.67GP-RTV and 6CB2CNT-RTV were used to fabricate the stretchable multimodal tactile sensors. Multimodal stretchable tactile sensors with dual-sensitive layers were fabricated by stacking the composites films, and the signals from the sensor units were sent out and analyzed separately. The stretchable multimodal tactile sensor in this study can simultaneously measure pressure and temperature.

As shown in Figure 6, the sensitive characteristics of the two sensor units were tested and recorded simultaneously. As shown in Figure 6a,b, the test was carried out under different temperature gradients, and the results showed that with increasing temperature, the sensitivity of 5.33CB2.67GP-RTV increased. The positive temperature effect of 5.33CB2.67GP-RTV was more noticeable than 6CB2CNT-RTV. The seven resistance-pressure curves of 6CB2CNT-RTV between 30 and 80 °C crossed each other in the low temperature range, while the resistance of 5.33CB2.67GP-RTV increased steadily with increasing temperature. No cross phenomenon was observed in the resistance-pressure curves of 5.33CB2.67GP-RTV. In addition, the curve for 90 °C was far from the other curves because the resistance increased sharply with increasing temperature.

As shown in Figure 6a, the GF values of the curves were 0.0198–0.0252 kPa^−1^, and the GF values of the curves shown in Figure 6b were 0.0018–0.0549 kPa^−1^. Compared with the stretchable tactile sensors with a single sensitive layer, the sensitivity of the 6CB2CNT-RTV sensor improved, while the GF of the 5.33CB2.67GP-RTV sensor fluctuated more than the 6CB2CNT-RTV sensor. This was due to the multilayer structure of the stretchable multimodal tactile sensor and the asymmetrical upper and lower arrangement of the electrodes. To observe the resistance changes in the stretchable multimodal tactile sensor under different stimuli, the 3D surfaces were used to illustrate the sensitivity characteristics of the stretchable multimodal tactile sensor, as shown in Figure 6d,e.

The temperature test results are shown in Figure 6c, which indicated better temperature sensitivity of 5.33CB2.67GP-RTV compared with 6CB2CNT-RTV. The GF values of 5.33CB2.67GP-RTV were 2.46%/°C (30–50 °C), 8.62%/°C (50–70 °C), 30.56%/°C (70–90 °C). The temperature sensitivity improved in each temperature range, which was possibly due to the multilayer structure, reduced heat loss. Additionally, 5.33CB2.67GP-RTV was located in the lower layer and thus closer to the heating table.

We found that the resistance of the stretchable multimodal tactile sensor was affected by both pressure and temperature, and the stretchable multimodal tactile sensor can effectively monitor pressure and temperature simultaneously. Thus, similar methods may be used to prepare stretchable tactile multimodal sensors that sense stretch and temperature simultaneously.

## 4. Signal Processing Circuit and Sensing Test

To further validate the compression and temperature monitoring functionality of the multimodal tactile sensors, a scene verification experiment was conducted. In this experiment, an Arduino UNO board was used to convert the resistance of the sensor units into a voltage signal, according to the voltage division principle. Afterward, resistance was calculated and output in real-time from the Arduino IDE to the PC, using a serial plotter. In the resistance test program, an Arduino IDE multithreading library scoop was used to establish the two-child thread algorithms to complete the simultaneous testing of the two sensor units. The formula for the voltage division principle was: *U*_out_ = (*U*_in_ × *R*_2_)/(*R*_1_ + *R*_2_), where *U*_in_ is the input voltage, *U*_out_ is the output voltage, *R*_1_ is the reference resistance, and *R*_2_ is the resistance to be tested. The input voltage *U*_in_ was set to 5 V and *R*_2_ was calculated based on the output voltage *U*_out_ and the known *R*_1_. Therefore, *R*_2_ was the real-time resistance of the sensor unit.

Figure 7a shows a photograph of the signal processing circuit of the stretchable multimodal tactile sensor, and Figure 7b shows the measured resistance based on the voltage division principle. The experimental signals were input into the PC through a USB cable, and then the electric signal curves were displayed on the PC in real-time. As shown in Figure 7c, the two real-time curves of the sensor units were observed in the window of the Arduino IDE serial plotter. The first peak was the change in the signal due to pressing of the sensor with a finger and then releasing it quickly. The second peak was the change in the signal when a cup filled with 90 °C hot water was placed on the surface of the sensor for 30 s and then moved away. The response amplitude of the two sensing units caused by temperature and pressure stimulation is different, and the response time of the two signals is significantly different. These measurement results further showed that our stretchable multimodal tactile sensor can simultaneously measure pressure and temperature and distinguish different stimuli. Therefore, this sensor holds potential for multi tactile parameters perception applications.

## 5. Conclusions

In this work, stretchable and flexible conductive rubber composites with force-sensitive and temperature-sensitive properties were fabricated, and we studied the mechanical and electrical properties of the conductive filler embedded with conductive rubber composites. The elastic modulus of the conductive rubber composites improved by 184% after embedding with conductive fillers, and the elongation at break and tensile strength both improved by 182% and 206%, respectively. Thus, our conductive rubber composites have potential for academic research and industrial applications as stretchable and flexible stretch, compression, and temperature sensors. Finally, a stretchable multimodal tactile sensor with high sensitivity and a large measurement range was manufactured.

The developed stretchable tactile sensor exhibited compression sensitivity values of 0.00204–0.01171 kPa^−1^ and stretch sensitivity of 2.63–37.45. The best temperature sensitivity values of the conductive rubber composites were 2.92%/°C (30–50 °C), 11.39%/°C (50–70 °C), and 19.14%/°C (70–90 °C). The test results of the stretchable multimodal tactile sensor showed that the temperature sensitivity results of the sensor unit with the best performance were 2.46%/°C (30–50 °C), 8.62%/°C (50–70 °C), and 30.56%/°C (70–90 °C). The results thus showed that the stretchable multimodal tactile sensor fabricated in this study can simultaneously monitor pressure and temperature. In addition, the signals were stable and did not interfere with each other, and the fabricated stretchable multimodal tactile sensor in the study did not require complex signal processing or a transmission circuit system.

In summary, conductive rubber composites were prepared by embedding hybrid conductive fillers, and a stretchable multimodal tactile sensor based on conductive rubber composites was developed via a simple fabrication process. The sensitive characteristics of the conductive rubber composites were tuned by embedding different carbon nanomaterials. The conductive composite stretchable multimodal tactile sensor exhibited high sensitivity and a wide measurement range for sensing pressure and temperature. Furthermore, the cyclic loading-unloading tests showed that the tactile sensor was reliable and durable. Therefore, this material can be used in various fields such as human-machine interfaces in intelligent automobiles, stretchable smart robots, and wearable medical devices.

## Figures and Tables

**Figure 1 polymers-14-01294-f001:**
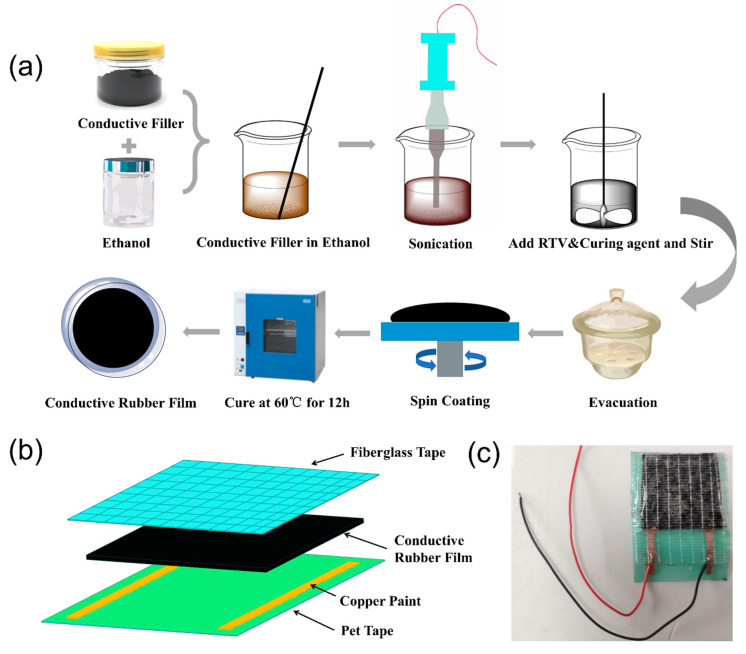
(**a**) Schematic of the conductive rubber composite fabrication process, (**b**) structure of the fabricated stretchable tactile sensor, and (**c**) photograph of the fabricated stretchable tactile sensor.

**Figure 2 polymers-14-01294-f002:**
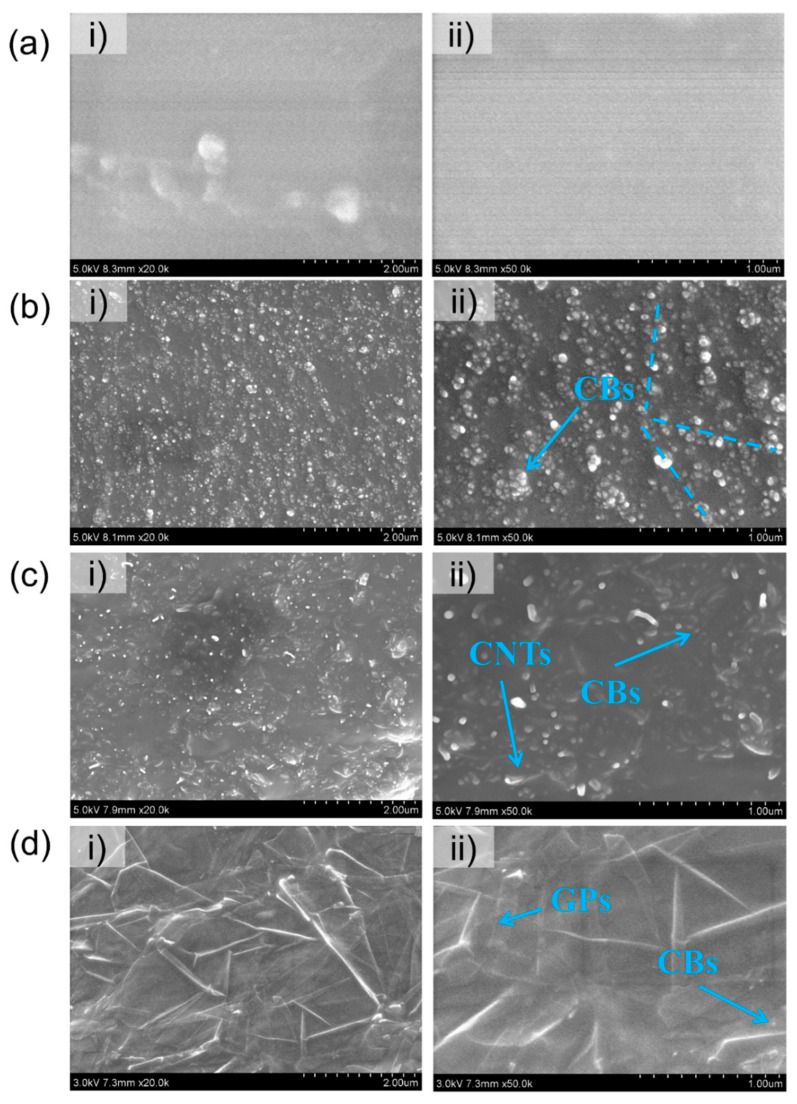
(**a**) SEM micrographs of RTV films of 20,000× (**i**) and 50,000× (**ii**) magnification, (**b**) SEM micrographs of CB-embedded composites films at 20,000× (**i**) and 50,000× (**ii**) magnification, (**c**) SEM micrographs of the CB/CNT-embedded composites films at 20,000× (**i**) and 50,000× (**ii**) magnification, (**d**) SEM micrographs of the CB/GP-embedded composites films at 20,000× (**i**) and 50,000× (**ii**) magnification.

**Figure 3 polymers-14-01294-f003:**
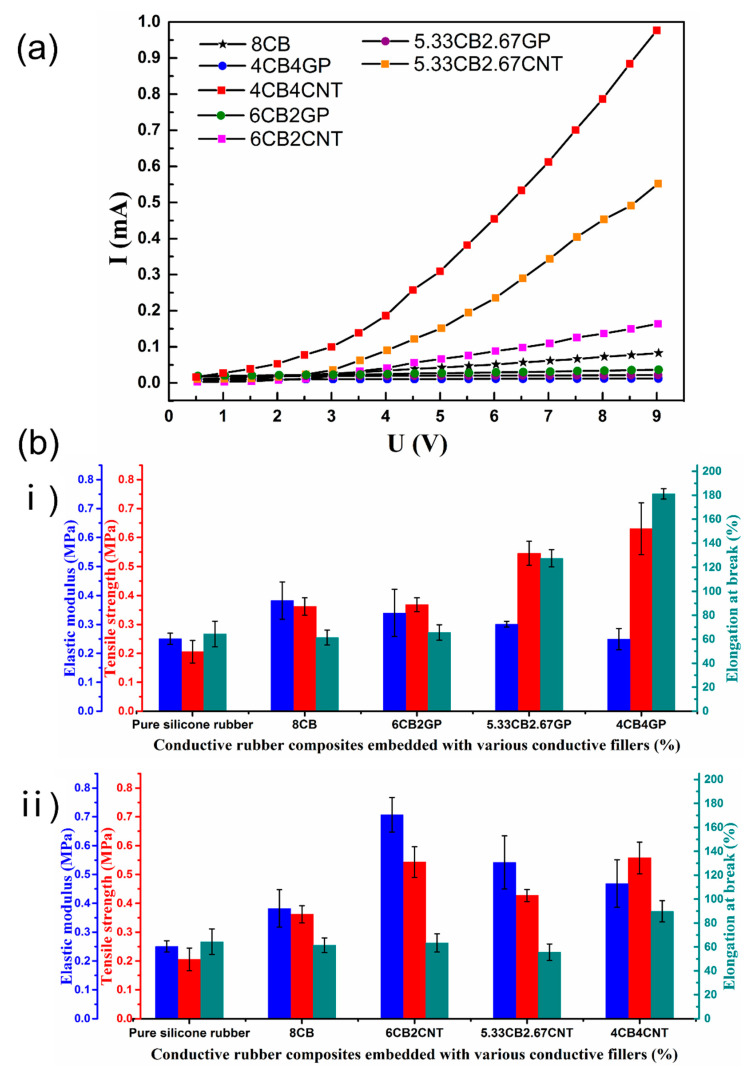
(**a**) Volt-ampere characteristics of the conductive rubber composites embedded with various conductive fillers, and (**b**) mechanical properties including the elastic modulus, tensile strength, and elongation at break of the conductive rubber composites embedded with CB/GP (**i**) and CB/CNT (**ii**).

**Figure 4 polymers-14-01294-f004:**
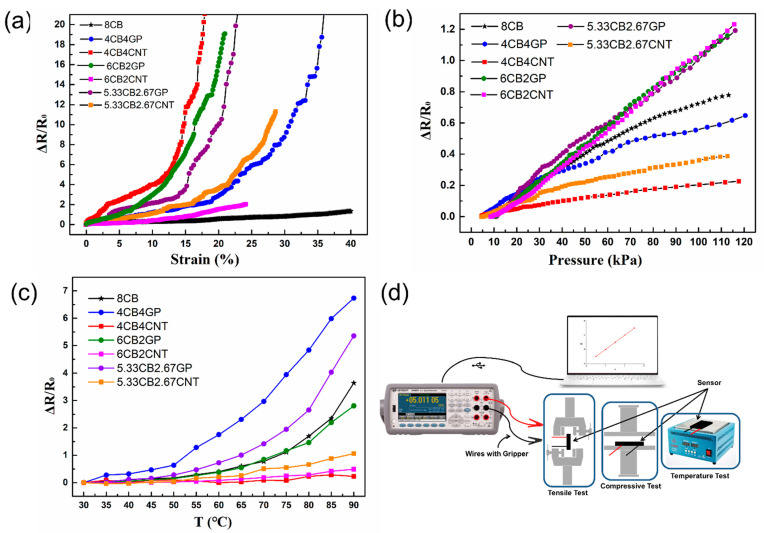
The test results and schematic of the test devices, showing the relative resistance changes in the conductive rubber composite stretchable tactile sensors with various material mass ratios as a function of strain (**a**) and pressure (**b**), and (**c**) relative changes in the resistance of the stretchable tactile sensors as a function of temperature, and (**d**) schematic of the test devices.

**Figure 5 polymers-14-01294-f005:**
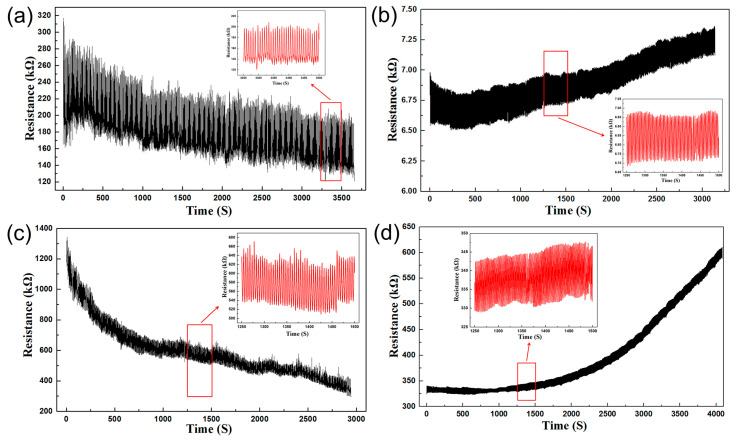
Repeatability tests of the stretchable tactile sensors based on 6CB2CNT-RTV under stretch (500 times) (**a**) and compression (1500 times) (**b**) cyclic loading-unloading. Repeatability testing of the stretchable tactile sensors based on 5.33CB2.67GP-RTV was conducted 500 times stretch (**c**) and 1500 times for compression (**d**) cyclic loading-unloading.

**Figure 6 polymers-14-01294-f006:**
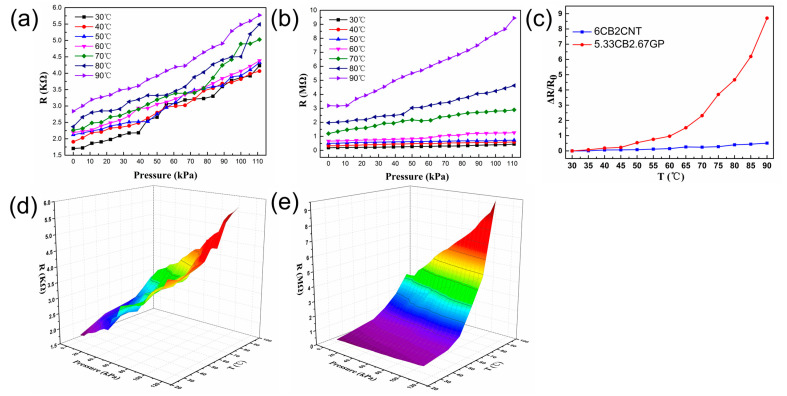
Changes in resistance of the 5.33CB2.67GP-RTV (**a**) and 6CB2CNT-RTV (**b**) sensor units as a function of pressure under different temperature gradients, (**c**) relative changes in resistance of the stretchable multimodal tactile sensor based on 5.33CB2.67GP-RTV and 6CB2CNT-RTV as a function of temperature, where the 3D surfaces were used to comprehensively illustrate the changes in resistance of the sensor units based on 5.33CB2.67GP-RTV (**d**) and 6CB2CNT-RTV (**e**).

**Figure 7 polymers-14-01294-f007:**
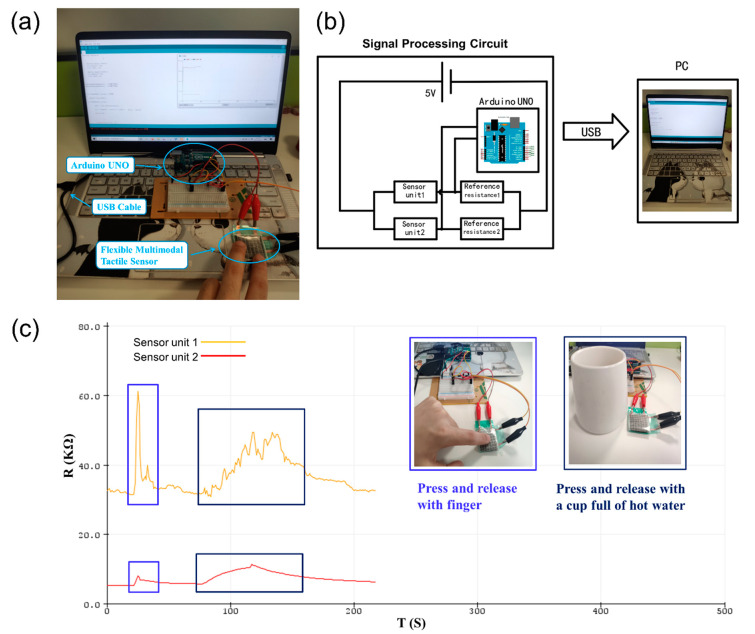
(**a**) Photograph of the signal processing circuit of the stretchable multimodal tactile sensor, (**b**) schematic of the signal processing circuit, and (**c**) real-time relative changes in resistance of the stretchable multimodal tactile sensor for monitoring finger pressing and pressing by a cup filled with hot water.

**Table 1 polymers-14-01294-t001:** Sensitivities of conductive rubber composites with different material ratios.

Conductive Rubber Composites	Temperature Sensitivity	Stretch Sensitivity	Compression Sensitivity
8CB-RTV	0.89%/°C (30–50 °C) 3.06%/°C (50–70 °C) 13.92%/°C (70–90 °C)	2.63 (0–40%)	0.803% kPa^−1^
6CB2CNT-RTV		4.54 (0–15%) 13.65 (15–25%)	1.171% kPa^−1^
6CB2GP-RTV		25.11 (0–10%) 129.81 (10–20%)	1.161% kPa^−1^
5.33CB2.67CNT-RTV		14.73 (0–20%) 84.56 (20–30%)	0.401% kPa^−1^
5.33CB2.67GP-RTV	1.39%/°C (30–50 °C) 5.62%/°C (50–70 °C) 19.91%/°C (70–90 °C)	20.39 (0–15%) 115.99 (15–20%)	1.097% kPa^−1^
4CB4CNT-RTV		37.45 (0–12%) 247.54 (12–20%)	0.204% kPa^−1^
4CB4GP-RTV	2.92%/°C (30–50 °C) 11.39%/°C (50–70 °C) 19.14%/°C (70–90 °C)	12.03 (0–20%) 81.68 (20–35%)	0.577% kPa^−1^

## Data Availability

The data used to support the findings of this study are available from the corresponding author upon request.

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
