# Peer review of "Highly Stretchable and Sensitive Multimodal Tactile Sensor Based on Conductive Rubber Composites to Monitor Pressure and Temperature"

_polymers, 2022, doi:10.3390/polym14071294_

Round 1

Reviewer 1 Report

The authors present a manuscript describing the possibility to create a sensor of pressure and temperature made by conductive carbon materials. Even if the work is not a breakthrough in science and many of these kind of sensors have been already done and presented in literature (https://www.sciencedirect.com/science/article/pii/S2452177917300178 ; https://pubs.acs.org/doi/10.1021/acs.chemrev.8b00340 etc ) the manuscript is well done. The introduction is complete and the experimental part is very well done, presenting both many references and all the details of the experimental procedure. The discussion is well completed with graphs and images, as well as the conclusions.

Reviewer 2 Report

Comments to the author

Dear Authors,

This study well investigates the stretchable conductive rubber for pressure and temperature sensors. This is an important study to be published after major revision as follows

  1. Correct the Figure 2 with numbering.
  2. “According to a previous study, the percolation threshold of the carbon-based conductive filler was approximately 5–8%.” Is it volume or weight percent? Please put reference.
  3. Why graphene (GP) less able to disperse in RTV rubber? (page 9)
  4. “stretch strength” should be written as “tensile strength” or “modulus” at a particular elongation.
  5. Why the regular increase in the elongation at break values for CB:GP systems compared to CB:CNT systems was observed? Little more discussion on the mechanical properties should be discussed. Author may see these paper [(a) L. Valentini, S. Bittolo Bon, M.A. Lopez-Manchado, R. Verdejo, L. Pappalardo, A. Bolognini, A. Alvino, S. Borsini, A. Berardo, N.M. Pugno, Synergistic effect of graphene nanoplatelets and carbon black in multifunctional EPDM nanocomposites, Composites Science and Technology, Volume 128, 2016, Pages 123-130. (b) Alam, M.N., Kumar, V., Potiyaraj, P. et al.Mutual dispersion of graphite–silica binary fillers and its effects on curing, mechanical, and aging properties of natural rubber composites.  Bull. (2021). https://doi.org/10.1007/s00289-021-03608-x.]
  6. In Figure 4a and 4b the y-axis scale may be ΔR/R0. Please check and modify.
  7. Color and symbols for different mixes should be same in all figures so that the behavior of the composites can be easily compared.
  8. In Figure 5, changes in the resistance with cycles are not a good sign for a strain sensor with practical utility. Author should include the maximum physical stability with number of cycles. Such behavior could be due to the plastic deformation. Please do the dynamic mechanical analysis to find out the mechanism.
  9. In compressive repeatability test a decrease in the resistance up to certain cycles and then it again increases. What is the reason behind the result?
  10. Please provide how much time is needed to fully sense a temperature and the relaxation time to come back at the ground stage (repeatability test for temperature sensing is needed).

Reviewer 3 Report

This is a very interesting manuscript, containing description of experimental results concerning preparation of:

1. conductive silicone rubber (SR) composites, filled with different kinds of carbon-based conductive substrates: CB, GP, and CNTs;

2. stretchable tactile sensors

and studies of their electrial and mechanical properties.

Obtained results are important and may find practical applications for fabrication of electronic devices which could be useful for measurements of pressure and temperature.

Only minor corrections of this manuscript are required.

  1. What kind of silicone RTV rubber and silicone crosslinking agent were applied ?
  2. Was it addition (hydrosilylation) type of silicone rubber or condensation type ?
  3. If possible, I would be satisfied, if authors could provide molecular characterisctics of SR: Mw, Mn, and polydispersity of molecular weights: Mw/Mn.
  4. Why only 8 % content of carbon fillers (with respect to silicone rubber matrix) was used ? Is it optimal content of carbon fillers ?

-

Round 2

Reviewer 2 Report

Thank you very much for your corrections and hopeful to get more advanced studies in future regarding temperature sensing. The paper is now acceptable based on my suggestion.